# TMA, A Forgotten Uremic Toxin, but Not TMAO, Is Involved in Cardiovascular Pathology

**DOI:** 10.3390/toxins11090490

**Published:** 2019-08-26

**Authors:** Kinga Jaworska, Dagmara Hering, Grażyna Mosieniak, Anna Bielak-Zmijewska, Marta Pilz, Michał Konwerski, Aleksandra Gasecka, Agnieszka Kapłon-Cieślicka, Krzysztof Filipiak, Ewa Sikora, Robert Hołyst, Marcin Ufnal

**Affiliations:** 1Department of Experimental Physiology and Pathophysiology, Laboratory of Centre for Preclinical Research, Medical University of Warsaw, 02-106 Warsaw, Poland; 2Department of Hypertension and Diabetology, Medical University of Gdansk, 80-211 Gdansk, Poland; 3Laboratory of Molecular Bases of Aging, Nencki Institute of Experimental Biology, Polish Academy of Sciences, 02-093 Warsaw, Poland; 4Department of Soft Condensed Matter, Institute of Physical Chemistry, Polish Academy of Sciences, 01-224 Warsaw, Poland; 51st Chair and Department of Cardiology, Medical University of Warsaw, 02-106 Warsaw, Poland

**Keywords:** trimethylamine, TMAO, bacterial metabolites, biomarkers

## Abstract

Trimethylamine-N-oxide (TMAO) has been suggested as a marker and mediator of cardiovascular diseases. However, data are contradictory, and the mechanisms are obscure. Strikingly, the role of the TMAO precursor trimethylamine (TMA) has not drawn attention in cardiovascular studies even though toxic effects of TMA were proposed several decades ago. We assessed plasma TMA and TMAO levels in healthy humans (HH) and cardiovascular patients qualified for aortic valve replacement (CP). The cytotoxicity of TMA and TMAO in rat cardiomyocytes was evaluated using an MTT test. The effects of TMA and TMAO on albumin and lactate dehydrogenase (LDH) were assessed using fluorescence correlation spectroscopy. In comparison to HH, CP had a two-fold higher plasma TMA (*p* < 0.001) and a trend towards higher plasma TMAO (*p* = 0.07). In CP plasma, TMA was inversely correlated with an estimated glomerular filtration rate (eGFR, *p* = 0.002). TMA but not TMAO reduced cardiomyocytes viability. Incubation with TMA but not TMAO resulted in the degradation of the protein structure of LDH and albumin. In conclusion, CP show increased plasma TMA, which is inversely correlated with eGFR. TMA but not TMAO exerts negative effects on cardiomyocytes, likely due to its disturbing effect on proteins. Therefore, TMA but not TMAO may be a toxin and a marker of cardiovascular risk.

## 1. Introduction

Recently, trimethylamine N-oxide (TMAO) has gained a lot of attention as a marker of cardiovascular risk. However, data are somehow contradictory [1,2,3,4,5,6,7,8,9,10].

Plasma TMAO originates from the liver, which oxidizes trimethylamine (TMA), a gut bacteria product, to TMAO. In addition, plasma TMAO increases after the ingestion of dietary TMAO, e.g., saltwater fish and seafood [11]. 

Though the number of publications on TMAO are rapidly growing—only in the last three-to-four years, over 600 papers have been published according to PubMed—the mechanisms linking increased plasma levels of TMAO to cardiovascular diseases remain unknown [12]. Furthermore, conflicting results exist whether TMAO exerts beneficial or harmful effects on living organisms [13,14,15,16]. Previously, we demonstrated that chronic, low-dose oral TMAO treatment exerted positive effects on a pressure-overloaded heart in rats and was associated with a decrease in the plasma level of TMA, suggesting that not only TMAO but also TMA may exert biological effects on the circulatory system [17]. Very recently, we found that TMA increased arterial blood pressure in rats [18] and that ageing, a risk factor for cardiovascular diseases, was associated with increased plasma TMA in rats. The latter, but not TMAO, exerted a toxic effect on vascular smooth muscle cells (VSMCs). Specifically, TMA affected VSMCs viability at concentrations only two-to-three-fold higher than those in portal blood in rats. In contrast, TMAO did not exert cytotoxic effects at concentrations exceeding its physiological level by 1000-fold [19]. 

Notably, the toxic effect of TMA was described in several clinical and experimental papers in the early 20th century [20,21]. Furthermore, TMA was suggested to be a uremic toxin as far back as 1981 [22]. With this regard, it has been well-established that the accumulation of uremic toxins is involved in the aetiology of cardiorenal syndrome [23,24,25]. 

Strikingly, the plasma level of TMA has not gained much attention in recent cardiovascular studies and, to the best of our knowledge, no clinical studies have evaluated the association between plasma TMAO levels and cardiovascular diseases in addition to the measurements of plasma TMA levels. This may result from the fact that papers showing the toxic effect of TMA are not available in widely-used medical databases that collect medical literature from recent decades. 

Therefore, our study sought to examine the plasma levels of TMA and TMAO in healthy humans and in cardiovascular patients. Our previous experimental work in rats showed an association between lower plasma TMA and the improved function of pressure-overloaded hearts [17]. Therefore, we investigated patients with a pressure-overloaded left ventricle which were qualified for aortic valve replacement due to severe aortic stenosis (AS). Furthermore, to evaluate a potential involvement of the molecules in cardiovascular pathology, we examined the effect of TMA and TMAO on cardiomyocytes, cardiac lactate dehydrogenase (LDH) and albumin protein structure.

## 2. Results

### 2.1. Plasma TMA and TMAO in Healthy Humans and Cardiovascular Patients

The plasma TMA level was over two-fold higher (*p* < 0.001) in cardiovascular patients in comparison to the healthy group. There was also a trend (*p* = 0.07) towards a higher plasma TMAO level in this group (Table 1). 

In cardiovascular patients, there was a significant inverse correlation between plasma TMA and estimated glomerular filtration rate (eGFR) (r = −0.66, *p* = 0.002), whereas TMAO levels only tended to be inversely correlated with eGFR (r = −0.43, *p* = 0.065), (Figure 1).

### 2.2. The Effect of TMA and TMAO on Cardiomyocytes Viability (MTT Assay)

TMA at 10 mmol/L and 1 mmol/L concentrations significantly reduced cell viability after 24 and 72 hours of treatment, respectively. In contrast, TMAO at an equimolar concentration had no effect on cardiomyocytes viability (Figure 2).

MTT findings were confirmed by a morphological observation of TMA or TMAO-treated cells (Figure 3). 

TMA concentrations of 10 and 100 mmol/L was cytotoxic, causing cell shrinkage and their detachment from the bottom of the plate. In contrast, TMAO at the same concentrations did not influence cell morphology. However, it lowered the density of cell culture, indicating that cell proliferation but not survival was affected. Notably, a concomitant treatment with TMAO (100 mmol/L) protected cardiomyocytes against the deleterious effect of TMA (10 mmol/L), i.e., a significant increase of cardiomyocyte viability was observed (Figure 2) and cells maintained normal morphology even after 10 days of treatment (Figure 4), indicating a long-lasting protective effect of TMAO.

### 2.3. The Effect of TMA and TMAO on Protein Structure

Twenty four-hour incubation with TMA resulted in the degradation of the protein structure of albumin and LDH. Specifically, there was a six-fold and three-fold increase in the diffusion coefficient (D) for albumin and LDH, respectively, at TMA concentrations of 100 and 200 mmol/L. In contrast, incubation with TMAO did not affect the D for albumin and LDH (Figure 5). 

## 3. Discussion

The novel finding of our study is that cardiovascular patients showed an increased plasma TMA level that was inversely correlated with GFR. Furthermore, we found that TMA but not TMAO exerted biological effects which may be detrimental for the circulatory system. 

Plasma TMAO originates from the liver oxygenation of TMA, a gut bacteria product of choline and carnitine [26]. The second source of plasma TMAO is dietary TMAO from seafood and fish [11]. Therefore, the plasma TMAO concentration depends on numerous factors including diet, the intestinal absorption of TMA and TMAO, the liver oxidation of TMA to TMAO, and the excretion of TMA and TMAO by the kidneys. 

A number of clinical studies have shown an association between elevated plasma TMAO levels and increased cardiovascular risk. However, data are somehow inconsistent [1,2,3,4,5,6,7,8,9,10], and the underlying mechanisms remain unclear [12]. 

Interestingly, none of the above-mentioned studies assessed the plasma level of TMA in addition to TMAO measurements. This is striking, as numerous studies have shown the toxic effects of TMA. In humans, eye [21] and skin [27] irritation have been shown after exposure to TMA. Apart from that, reproductive/developmental toxicity has been reported [28]. We found that TMA but not TMAO exerted a toxic effect on VSMCs [19], and a conference paper reported that TMA but not TMAO exerts vasoconstriction [29].

In fact, due to the obvious toxicity and, at the same time, widespread use in industry, various exposure limit guidelines with a detailed description of toxicity are available such as “Recommendation from the Scientific Committee on Occupational Exposure Limits” by the European Union Commission [30]. Finally, it was already suggested that TMA is a uremic toxin forty years ago [22]. 

Here, we found that plasma TMA is more than twice as high in cardiovascular patients compared to healthy individuals, which was accompanied by a less pronounced difference in TMAO levels. Our study also shows that kidney deterioration plays a key role in the increased plasma level of TMA, as demonstrated by a significant inverse correlation between the plasma level of TMA and eGFR. 

Uremic cardiomyopathy is a cardiac pathology found in patients with chronic kidney disease. Though both TMA and TMAO are increased in kidney failure [22,31], here, we found that only TMA exerted negative effects on cardiomyocytes. Specifically, we found that TMA but not TMAO decreased the proliferation and viability of cardiomyocytes. Moreover, the concomitant treatment with TMAO protected cardiomyocytes against the cytotoxic effect of TMA. The protective action of TMAO was evident after 10 days of treatment.

Our findings suggest that the toxic effect of TMA on cardiomyocytes may be due to a triggering role of TMA in the degradation of proteins structure. Specifically, we found that after 24 hours of incubation with TMA but not TMAO, the protein structure of cardiac LDH and albumin was degraded. This was evident at a TMA concentration of 100 mmol/L. Though such concentration significantly exceeded TMA plasma levels in humans, it is very likely that smaller conformational changes of proteins that alter their biological functions may occur at lower TMA concentrations. This assumption is supported by our experiments with cardiomyocytes and other studies showing the toxic effect of TMA at relatively low concentrations [20,21,28,32].

With regard to a potentially protective role of TMAO, there are numerous other studies showing that TMAO protects the structural and functional proteins of cells from denaturants such as high osmotic pressure, hydrostatic pressure, NaCl, urea or a high temperature [15,33,34,35].

It has been well-established that a decreased GFR is an independent cardiovascular risk factor [36,37]. Here, we found that cardiovascular patients had an increased plasma TMA level, which was associated with a decreased eGFR. An increased plasma TMA was also reported in kidney failure patients in other studies [38,39], providing evidence for the important role of renal deterioration in TMA plasma level increase.

Our study shows that, at equimolar concentrations, TMA but not TMAO exerts a cytotoxic effect on cardiomyocytes, perhaps due to the disturbing effect of TMA on cardiac LDH and, likely, other proteins. Finally, numerous studies published several decades ago showed the toxic effect of TMA on the respiratory tract [31], the reproductive system [28], eye [21] and skin [27].

Therefore, we propose that it is TMA and not TMAO that may be a marker and mediator of cardiovascular diseases. TMA may be one of the links between the progressive deterioration of kidney and heart functions, as well as mediator in cardiorenal syndrome.

Given that in our experiments, TMA degraded the protein structure of two very different functional proteins, i.e., albumin and cardiac LDH, and exerted a toxic effect on cardiomyocytes, it is most likely that the circulatory effects of TMA are very nonspecific, i.e., involve a disturbing effect of TMA on the protein structure of numerous functional proteins (receptors and transporters of mediators), rather than one specific pathway.

A limitation of our study is that we evaluated a relatively small and distinct group of cardiovascular patients, i.e., patients with aortic stenosis. In addition, further experimental studies are needed to elucidate the involvement of TMA in pathological processes underlying cardiovascular diseases.

## 4. Conclusions

Our study demonstrates that cardiovascular patients show increased plasma TMA which is inversely correlated with the eGFR. Furthermore, we found the TMA but not TMAO exerts cytotoxic effect on cardiomyocytes, likely due to the disturbing effects of the molecule on cardiac LDH and other proteins’ structures. Therefore, our findings strongly suggest that TMA but not TMAO may be a marker and mediator of cardiovascular diseases. Future clinical studies determining the association between cardiovascular risk and plasma TMAO should obligatorily evaluate plasma TMA levels.

## 5. Materials and Methods 

### 5.1. Plasma TMA and TMAO Level in Healthy Humans and Cardiovascular Patients

Written informed consent was obtained from all subjects. The study conformed to the standards set by the latest revision of the Declaration of Helsinki and was approved by the Bioethical Committee of the Medical University of Warsaw (permissions no: KB/211/2018, 19 November 2018, and KB/4/2019, 14 January 2019).

The study group included patients with severe AS who qualified either for surgical aortic valve replacement or transcatheter aortic valve implantation recruited from the Public Central Teaching Hospital of the Medical University of Warsaw. The control group consisted of healthy subjects recruited from the GP practice “Zdrowie” in Warsaw. The inclusion criteria for healthy subjects was age 18–99, and the exclusion criteria were the following: A medical history of chronic diseases, current therapy with any medications, antibiotic treatment within the last one month, acute gastrointestinal disease in the last 4 weeks, and the use of dietary supplements in the last 7 days. The inclusion criteria for AS patients were age 18–99 years, severe AS defined as aortic valve area (AVA) < 1.0 cm^2^ and/or indexed AVA of < 0.6 cm^2^/m^2^ as calculated by the continuity equation on transthoracic echocardiography (both patients with high gradient and low gradient severe AS were enrolled) irrespective of the presence of AS symptoms, qualified for interventional (surgical or transcatheter) treatment by the heart team, in accordance with the guidelines of the European Society of Cardiology [40,41]. Exclusion criteria in the AS group included: Heart failure attributable to causes other than severe AS, active neoplastic disease, chronic inflammatory disease, chronic or acute (within the last 4 weeks) gastrointestinal disease, antibiotic treatment in the last month, and the use of dietary supplements in the last 7 days. In AS patients, blood samples were collected during their preprocedural diagnostic work-up when they were clinically stable. The estimated glomerular filtration rate (eGFR) was calculated using the Modification of Diet in Renal Disease (MDRD) formula.

Blood was collected into precooled EDTA tubes and chilled in a refrigerator. Samples were centrifuged for 15 minutes at 2000× *g*, 60 min after blood collection. Plasma was frozen at −25 °C until assayed.

### 5.2. The Effect of TMA and TMAO on Cardiomyocytes In Vitro

#### 5.2.1. Cells and Treatment

Rat cardiomyocytes (H9c2 cell line) were purchased from ATCC (LGC Standards, Lomianki, Poland). Cells were cultured in DMEM (Sigma Aldrich, St.Louis, MO, USA), supplemented with 10% FBS (Biowest, Cytogen, Poland), L-glutamine and antibiotic and antymycotic solution. TMA and TMAO solutions were dissolved in a cell culture medium and diluted to obtain the desired concentrations.

#### 5.2.2. Metabolic Activity Assay in Cardiomyocytes (MTT Assay)

The MTT assay was used to check viability of cardiomyocytes treated with different concentrations of TMA and TMAO. Cells were seeded in a 96-well plate at a of density 1500 cells per well. After one day, the medium was exchanged for those containing TMA or TMAO. A 3-(4,5-Dimethylthiazol-2-yl)-2,5-diphenyltetrazolium bromide (MTT) (Sigma-Aldrich) solution (5 mg/mL) was added at indicated time points, and cells were incubated for 2 h at 37 °C in a humidified atmosphere (5% CO2). Formazan formed in living cells was dissolved in dimethyl sulfoxide (DMSO) (Sigma-Aldrich), and the absorbance of the solution was measured at 570 nm using a microplate reader (Reader 400 SFC, LabInstruments, Hamburg, Germany).

### 5.3. The Effect of TMA and TMA on Protein Structure 

We evaluated the effect of TMAO and TMA on bovine serum albumin (BSA) (Sigma-Aldrich) and cardiac lactic dehydrogenase (LDH) (Sigma-Aldrich) incubated with those agents at 20 °C for 24 h. The proteins were fluorescently labelled with Atto 488 dye (Atto-Tec GmbH, Siegen, Germany) via NHS-ester coupling reaction. Changes in protein structure were measured with the use of fluorescence correlation spectroscopy (FCS), which enables the determination of a fluorescent probe’s diffusion coefficient *D* related with the probe radius *r_p_* by the Stokes–Einstein relation. Protein unfolding (its increased hydrodynamic radius) results in a decrease in its *D* value. Otherwise, as the protein is degraded, the biomolecule diffusion coefficient increases. 

FCS measurements were performed using setup based on Nikon C1 confocal microscope (Nikon Instruments, Tokyo, Japan), fitted with a PicoQuant LSM upgrade kit (PicoQuant GmbH, Berlin, Germany). A PicoQuant pulsed diode laser of wavelengths of 485 nm, driven by a Sepia II module was used for fluorescence excitation. Data acquisition was controlled by SymPhoTime 64 software (Berlin, Germany).

### 5.4. Evaluation of TMA and TMAO Plasma Concentrations

The plasma concentrations of TMA and TMAO were evaluated using a Waters Acquity Ultra Performance Liquid Chromatograph coupled with a Waters TQ-S Triple-Quadrupole Mass Spectrometer. The mass spectrometer operated in the multiple-reaction monitoring (MRM)-positive electrospray ionization (ESI) mode, as we have previously described [42]. 

### 5.5. Data Analysis and Statistics

Differences between the groups were assessed using the t-test. The relationships between the concentration of TMA/TMAO and investigated variables were determined by Pearson’s correlation coefficient. A value of two-sided *p* < 0.05 was considered significant. Analyses were conducted using Dell Statistica, version 13 (Dell Inc, Tulsa, OK, USA).

## Figures and Tables

**Figure 1 toxins-11-00490-f001:**
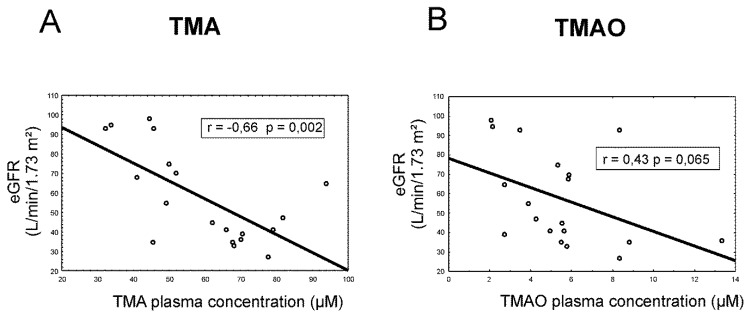
Correlation of plasma (**A**) TMA and (**B**) TMAO levels (µmol/L) with estimated glomerular filtration rate (eGFR) (mL/min/1.73 m²). r—Pearson correlation coefficient; TMA—trimethylamine; and TMAO—trimethylamine N-oxide.

**Figure 2 toxins-11-00490-f002:**
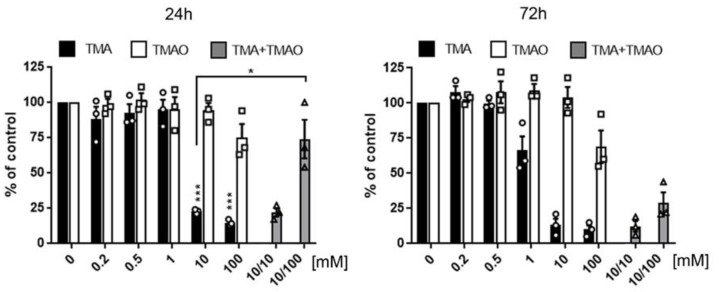
The influence of TMA and TMAO on cardiomyocytes in vitro. Comparison of the viability of cardiomyocytes (H9c2) treated with TMA or TMAO at increasing concentrations from 0.1 to 100 mmol/L and concomitant treatment with TMA and TMAO, 10/10 (10 mmol/L TMA/10 mmol/L TMAO), 10/100 (10 mmol/L TMA/100 mmol/L TMAO). An MTT test was performed after 24 and 72 h of treatment, measurement was performed in three independent experiments, and results for each experiment were normalized to the control (untreated) cells; graphs show mean ± SE, * *p* < 0.05, *** *p* < 0.001, by a t-test. TMA—trimethylamine; TMAO—trimethylamine N-oxide.

**Figure 3 toxins-11-00490-f003:**
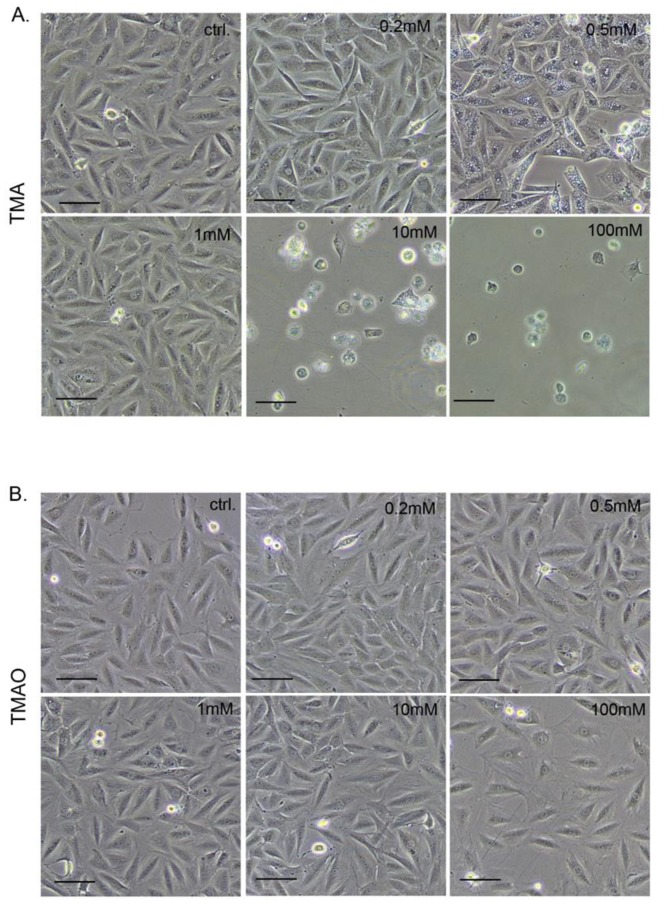
Representative pictures showing changes in the number and morphology of cells after 72 hours of treatment with (**A**) TMA and (**B**) TMAO—bar 100 µm. ctrl.—control; TMA—trimethylamine; and TMAO—trimethylamine N-oxide.

**Figure 4 toxins-11-00490-f004:**
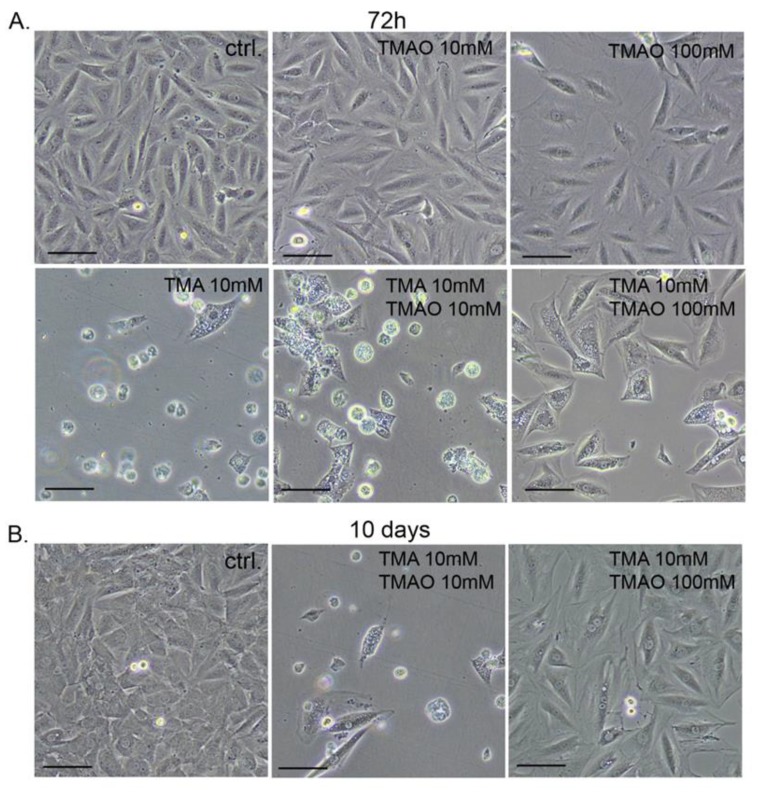
Representative pictures showing morphology of cells after (**A**) 72 hours and (**B**) 10 days of treatment with TMAO or TMA or the mixture of both compounds—bar 100 µm. ctrl.—control; TMA—trimethylamine; and TMAO—trimethylamine N-oxide.

**Figure 5 toxins-11-00490-f005:**
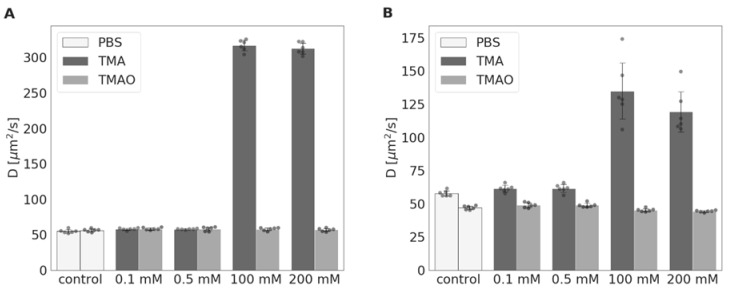
Diffusion coefficient obtained for (**A**) albumin and (**B**) LDH after 24-hour incubation with either PBS (control), TMA, or TMAO at increasing concentrations (n = 6). With protein degradation, the biomolecule diffusion coefficient increases. D—diffusion coefficient; LDH—lactate dehydrogenase; PBS—phosphate-buffered; TMA—trimethylamine; and TMAO—trimethylamine N-oxide.

**Table 1 toxins-11-00490-t001:** Demographics and plasma level of TMA and TMAO in the healthy group and in patients with severe aortic stenosis (AS). TMA—trimethylamine, TMAO—trimethylamine N-oxide.

Characteristic	Healthy Group (n = 9)	AS Group (n = 19)	*p*-Value
Ethnicity (Caucasian/other)	9/0	19/0	-
Male/Female	5/4	11/8	-
Age (years)	38.9 ± 4.8	74.5 ± 2.3	<0.001
eGFR (mL/min/1.73 m²)	101.5 ± 6.3	57.4 ± 5.45	<0.001
Plasma TMA (µmol/L)	23.2 ± 2.1	59.5 ± 3.9	<0.001
Plasma TMAO (µmol/L)	3.6 ± 0.4	5.5 ± 0.6	0.07

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
