# Peer review of "TMA, A Forgotten Uremic Toxin, but Not TMAO, Is Involved in Cardiovascular Pathology"

_toxins, 2019, doi:10.3390/toxins11090490_

Round 1

Reviewer 1 Report

The authors determined the plasma level of TMA and TMAO in healthy subjects and patients qualified for an aortic valve replacement with severe aortic stenosis and further evaluated the correlation of plasma TMA and TMAO levels with eGFR. Their results revealed that Plasma TMA of patients was significantly increased and inversely correlated with estimated glomerular filtration rate (eGFR). Furthermore, they also determined the effect of TMA and TMAO on cardiomyocytes viability and indicated that TMA reduced cardiomyocytes viability and released necrotic protein such as LDH. Although these findings are interesting, but is also preliminary for the publication. They may need to pay further attention to the following concerns:

Vascular smooth muscle cells (VSMCs) are the major cell type in the aorta and their main function is to regulate blood flow and pressure through vessel wall contraction and relaxation which play an important role in controlling cardiovascular function. The authors may investigate the role of TMA and TMAO in VSMC pathogies such as inflammation, migration, and proliferation.

Author Response

Dear Reviewer, thank you very much for agreeing to review our paper and your valuable comment.

We fully agree with the Reviewer that vascular smooth muscle cells (VSMCs) are the major cell type which play an important role in controlling cardiovascular function.

In fact, we have investigated the effect of TMA and TMAO on VSMCs in a paper which was accepted for publication after we had submitted the present manuscript.

As the above-mentioned paper has just been made available online (Jaworska, Konop et al. 2019) we have added discussion on the effect of TMA and TMAO on VSMCs to the present manuscript, please see lines 40-44.

In general, the study looked at gut bacteria and TMAO/TMA in young vs old rats. In the study we found that in comparison to younger rats older rats showed increased plasma TMA which was associated with compromised kidney function (increased plasma creatinine and decreased renal creatinine clearance) which is in line with the present findings in humans. We also found that older rats had increased gut-to-blood penetration of TMA, which we were not able to evaluate in humans in this study group. Therefore, these two papers provide strong evidence that deterioration of kidney function (that is usually associated with both age and/or cardiovascular diseases) increases plasma TMA.

Regarding the effect of TMA and TMAO on VSMCs we found that TMA reduced cell viability after 24h of treatment. In contrast, TMAO at the same concentration had no impact on the cells even after 72 h of treatment. Importantly, treatment with TMAO at high concentrations 100mmol/L was not cytotoxic whereas TMA at the same concentration killed cells within 24h. The MTT findings were confirmed by a morphological observation of TMA or TMAO-treated cells. TMA caused cell shrinkage and detachment from the bottom of the plate while TMAO used at the same concentration did not influenced cell morphology.

Finally, we fully agree with the Reviewer that further experimental studies are needed to elucidate the involvement of TMA in pathological processes underlying cardiovascular diseases and we have added this to the limitation section, lines 175-177.

Nevertheless, to the best of our knowledge, our study is the first to show that cardiovascular patients demonstrate increased plasma TMA levels which is associated with deteriorated kidneys function (decreased eGFR), and that TMA but not TMAO exerts negative effects on cardiomyocytes, likely due to disturbing effect on proteins.

We believe that our findings may change the present paradigm for looking at TMAO as a cardiovascular risk factor. Our study suggests that an increased cardiovascular risk in patients with increased plasma TMAO level that has been widely reported, may depend on increased plasma level of its precursor i.e. TMA. Therefore, future clinical studies should obligatory evaluate both TMAO and TMA.

We hope that our present manuscript is worth publishing due to its clinical importance.

Refrence:

Jaworska, K., M. Konop, T. Hutsch, K. Perlejewski, M. Radkowski, M. Grochowska, A. Bielak-Zmijewska, G. Mosieniak, E. Sikora and M. Ufnal (2019). "TMA but not TMAO increases with age in rat plasma and affects smooth muscle cells viability." The Journals of Gerontology: Series A.

Reviewer 2 Report

1 A brief summary
   The paper describes that TMA but not TMAO exerts negative effects on cardiomyocytes, likely due to disturbing effect on proteins. TMA but not TAMO may be a toxin and a marker of cardiovascular risk.

   This is a very nice paper. However, I have some comments.

2 Overall evaluation
   The findings from this paper are excellent and novel regardless of a review.

3 Main problem
   This manuscript contained some questions described below.
   I think this paper has very interesting, this study contributes to future's clinical medicine largely. I have some questions from a point of view of clinical medicine.

   In this paper the authors focused TMA but not TMAO, almost papers describes the toxic effect of TMAO, according to recent published paper (Nutrients 2019,11,1821) TMAO from food or as a metabolite of TMA acts negative effect on kidney, cardiovascular system and various organs. And TMAO usually leaves the body in the urine. So I think that TMAO can cause kidney damage than TMA. Conversely, if there is a kidney disorder, TMAO is
more likely to be related to eGFR than TMA.
How do you think that?

   I want to know that why does the authors select the AS patient? I understand from this paper that AS patients have the pressure overloaded heart, but in cases of the other heart and vessel disorder (for example LVH or ischemia or volume overloaded heart DCM and so on ) how does the TMA or TMAO move? Please let me know because it may be experimental data or a literature review.

Author Response

Reviewer #2:

Re: Dear Reviewer, thank you very much for your time to review our paper and encouraging comments. Please find below answers to your specific remarks.

Comments and Suggestions for Authors

1 A brief summary 
   The paper describes that TMA but not TMAO exerts negative effects on cardiomyocytes, likely due to disturbing effect on proteins. TMA but not TAMO may be a toxin and a marker of cardiovascular risk.

   This is a very nice paper. However, I have some comments. 

2 Overall evaluation 
   The findings from this paper are excellent and novel regardless of a review. 

3 Main problem 
   This manuscript contained some questions described below. 
   I think this paper has very interesting, this study contributes to future's clinical medicine largely. I have some questions from a point of view of clinical medicine.

   In this paper the authors focused TMA but not TMAO, almost papers describes the toxic effect of TMAO, according to recent published paper (Nutrients 2019,11,1821) TMAO from food or as a metabolite of TMA acts negative effect on kidney, cardiovascular system and various organs. And TMAO usually leaves the body in the urine. So I think that TMAO can cause kidney damage than TMA. Conversely, if there is a kidney disorder, TMAO is
more likely to be related to eGFR than TMA. How do you think that?

RE:

Thank you for this important, however, very difficult questions to answer. Therefore, please forgive me perhaps too lengthy answers to your comments.

Indeed, recently TMAO has attracted large attention after the publication of several clinical papers suggesting that increased plasma TMAO is associated with increased risk of adverse cardiovascular, that is highlighted in the above-mentioned review in Nutrients, that we have now cited in the corrected manuscript, line 35.  

However, several clinical studies do not back the assumption(Fukami, Yamagishi et al. 2015, Yin, Liao et al. 2015, Meyer, Benton et al. 2016) or show that the association is dependent on race(Shafi, Powe et al. 2017), or show high intra-individual variation of plasma TMAO level over time(Kuhn, Rohrmann et al. 2017).

Most importantly, there are no interventional studies showing that manipulation of plasma TMAO level affects cardiovascular risk. Therefore, although there is some evidence that increased plasma TMAO is associated with increased cardiovascular risk, there is no evidence that this association is of causative nature.

Our present study strongly suggests that this is TMAO precursor i.e. TMA and not TMAO that is deleterious for the circulatory system. Therefore, TMAO is rather a lurking variable.

Actually, our hypothesis that this is TMA and not TMAO that is deleterious for the circulatory system may also be indirectly supported by above-cited clinical studies showing a positive association between elevated levels of plasma TMAO and an increased cardiovascular risk. Although those studies did not evaluate plasma TMA, it is very likely that increased plasma TMAO levels were accompanied by increased level of its precursor i.e. TMA

Finally, we are aware that several experimental studies show negative effect of TMAO. However, the discrepancy between our present findings and studies showing a negative effect of TMAO on the heart may result from several factors, such as tested doses of TMAO and experimental settings. For example, Makrecka-Kuka et al. showed that increased plasma TMAO impairs pyruvate and fatty acid oxidation in cardiac mitochondria in mice(Makrecka-Kuka, Volska et al. 2017). However, in the latter study, TMAO treatment was associated with a 22- to 23-fold increase in the plasma TMAO level. Finally, Organ et al. found that HF severity was significantly enhanced in transverse aortic constricted mice fed a TMAO-rich diet, which increased plasma TMAO by 16- to 17-fold(Organ, Otsuka et al. 2016). Therefore, physiological relevance of such findings is uncertain.

It is possible that very high doses of TMAO may be toxic, as most of biologically active molecules may exert toxic effect at high concentrations. But such effect is rather suprapharmacological phenomenon and is very unlikely to be present under physiological or pathological conditions which are characterized by 2-3 fold increase in plasma TMAO, as in cardiovascular patients(Tang, Wang et al. 2013).

In addition, our hypothesis that this is TMA and not TMAO that is toxic may also be supported by the following indirect evidence.

Firstly, plasma TMAO originates not only from TMA, a gut bacterial metabolite, but also from dietary TMAO i.e. TMAO-rich sea food(Cheung, Keski-Rahkonen et al. 2017). With this regard, it is accepted that humans consuming fish have lower risk of age-associated diseases such as cardiovascular and metabolic diseases(Widmer, Flammer et al. 2015, Schmedes, Balderas et al. 2018).

Secondly, numerous biophysical studies show that TMAO exerts positive effects i.e. protects cell proteins from denaturants such as high osmotic pressure, hydrostatic pressure, NaCl, urea or a high temperature(Zou, Bennion et al. 2002, Jethva and Udgaonkar 2018, Vigorita, Cozzolino et al. 2018). The importance of TMAO has been recognized in marine animals which are exposed to high hydrostatic stress (deep water) and osmotic stress (salty water)(Yancey, Speers-Roesch et al. 2018).

Thirdly, liver oxidizes TMA to TMAO. Since one of major functions of the liver is detoxification it seems plausible that toxic TMA is converted to nontoxic TMAO.

Finally, in contrast to neutrally smelling TMAO, TMA has a strong unpleasant smell of decaying fish. In this regard, the sense of smell has evolved to alert animals to the poisons and to avoid ingestion of foods that are harmful(Reed and Knaapila 2010).

Regarding your question about the effect of kidney on plasma TMAO vs TMA level it is very difficult to answer. This is because of the following:

Plasma TMAO will depend on plasma TMA level (TMA is oxidized by the liver to TMAO), the higher plasma TMA the more TMAO. Kidneys remove both TMA and TMAO. In addition, plasma TMAO will depend on a diet e.g. consumption of seafood that is rich in TMAO.

Interestingly, according to WHO reports, age-dependent and cardiovascular mortality is lower in the Japanese population, comparing to the North American (WHO. Global Health Observatory data – life tables: Japan. http://apps.who.int/gho/data/?theme=main&vid=60820. 2018 , WHO. Global Health Observatory data – life tables: USA. http://apps.who.int/gho/data/?theme=main&vid=61780. 2018) and at the same time there are data showing significantly higher urine TMAO concentrations in Japanese than in  Americans(Dumas, Maibaum et al. 2006), which may be caused by very high TMAO intake (due to seafood, fish-rich diet) in Japanese population.

I want to know that why does the authors select the AS patient? I understand from this paper that AS patients have the pressure overloaded heart, but in cases of the other heart and vessel disorder (for example LVH or ischemia or volume overloaded heart DCM and so on ) how does the TMA or TMAO move? Please let me know because it may be experimental data or a literature review.

RE: In this pilot study we have chosen AS patients as this is a good model of pressure overloaded heart. We looked at pressure-overloaded heart, because previously we found that a moderate increase in plasma TMAO due to TMAO supplementation may be beneficial in animal model of pressure-overloaded heart(Huc, Drapala et al. 2018). i.e. we found that rats treated with TMAO had reduces plasma NH2-terminal pro-B-type natriuretic peptide and cardiac fibrosis.

Finally, ample evidence highlights protective functions of TMAO, including the stabilization of proteins and cells exposed to hydrostatic stresses, for example in fish or bacteria exposed to hydrostatic stress (deep water)(Zou, Bennion et al. 2002, Jethva and Udgaonkar 2018, Vigorita, Cozzolino et al. 2018, Yancey, Speers-Roesch et al. 2018). Hydrostatic stress is augmented in AS patients’ heart therefore we thought it may be a good start to evaluate TMA and TMAO in this group of patients.

Nevertheless, currently we look at TMA and TMAO levels in other groups of cardiovascular patients including those with pressure overloaded heart due to systemic hypertension but also in patients with pulmonary hypertension and pulmonary embolism.

In the corrected manuscript, we have added a paragraph that highlights the limitation of our paper i.e. that we evaluated relatively small and distinct group of cardiovascular patients (lines 174-175).

Regarding your question about TMA and TMAO in other heart and vessel disorder (LVH, ischemia or DCM), to the best of our knowledge, our study is the first to evaluate TMA level. So far, plasma TMA level did not draw attention in cardiovascular studies. In contrast, there is several papers evaluating plasma TMAO level in chronic heart failure(Troseid, Ueland et al. 2015), coronary artery disease(Tang, Wang et al. 2013, Li, Obeid et al. 2017), peripheral artery disease(Yin, Liao et al. 2015, Senthong, Wang et al. 2016) and hemodialysis patients(Shafi, Powe et al. 2017).

References

Cheung, W., P. Keski-Rahkonen, N. Assi, P. Ferrari, H. Freisling, S. Rinaldi, N. Slimani, R. Zamora-Ros, M. Rundle, G. Frost, H. Gibbons, E. Carr, L. Brennan, A. J. Cross, V. Pala, S. Panico, C. Sacerdote, D. Palli, R. Tumino, T. Kuhn, R. Kaaks, H. Boeing, A. Floegel, F. Mancini, M. C. Boutron-Ruault, L. Baglietto, A. Trichopoulou, A. Naska, P. Orfanos and A. Scalbert (2017). "A metabolomic study of biomarkers of meat and fish intake." Am J Clin Nutr 105(3): 600-608.

Dumas, M. E., E. C. Maibaum, C. Teague, H. Ueshima, B. Zhou, J. C. Lindon, J. K. Nicholson, J. Stamler, P. Elliott, Q. Chan and E. Holmes (2006). "Assessment of analytical reproducibility of 1H NMR spectroscopy based metabonomics for large-scale epidemiological research: the INTERMAP Study." Anal Chem 78(7): 2199-2208.

Fukami, K., S. Yamagishi, K. Sakai, Y. Kaida, M. Yokoro, S. Ueda, Y. Wada, M. Takeuchi, M. Shimizu, H. Yamazaki and S. Okuda (2015). "Oral L-carnitine supplementation increases trimethylamine-N-oxide but reduces markers of vascular injury in hemodialysis patients." J Cardiovasc Pharmacol 65(3): 289-295.

Huc, T., A. Drapala, M. Gawrys, M. Konop, K. Bielinska, E. Zaorska, E. Samborowska, A. Wyczalkowska-Tomasik, L. Paczek, M. Dadlez and M. Ufnal (2018). "Chronic, low-dose TMAO treatment reduces diastolic dysfunction and heart fibrosis in hypertensive rats." Am J Physiol Heart Circ Physiol 315(6): H1805-H1820.

Jethva, P. N. and J. B. Udgaonkar (2018). "The Osmolyte TMAO Modulates Protein Folding Cooperativity by Altering Global Protein Stability." Biochemistry 57(40): 5851-5863.

Kuhn, T., S. Rohrmann, D. Sookthai, T. Johnson, V. Katzke, R. Kaaks, A. von Eckardstein and D. Muller (2017). "Intra-individual variation of plasma trimethylamine-N-oxide (TMAO), betaine and choline over 1 year." Clin Chem Lab Med 55(2): 261-268.

Li, X. S., S. Obeid, R. Klingenberg, B. Gencer, F. Mach, L. Raber, S. Windecker, N. Rodondi, D. Nanchen, O. Muller, M. X. Miranda, C. M. Matter, Y. Wu, L. Li, Z. Wang, H. S. Alamri, V. Gogonea, Y. M. Chung, W. H. Tang, S. L. Hazen and T. F. Luscher (2017). "Gut microbiota-dependent trimethylamine N-oxide in acute coronary syndromes: a prognostic marker for incident cardiovascular events beyond traditional risk factors." Eur Heart J 38(11): 814-824.

Makrecka-Kuka, M., K. Volska, U. Antone, R. Vilskersts, S. Grinberga, D. Bandere, E. Liepinsh and M. Dambrova (2017). "Trimethylamine N-oxide impairs pyruvate and fatty acid oxidation in cardiac mitochondria." Toxicol Lett 267: 32-38.

Meyer, K. A., T. Z. Benton, B. J. Bennett, D. R. Jacobs, Jr., D. M. Lloyd-Jones, M. D. Gross, J. J. Carr, P. Gordon-Larsen and S. H. Zeisel (2016). "Microbiota-Dependent Metabolite Trimethylamine N-Oxide and Coronary Artery Calcium in the Coronary Artery Risk Development in Young Adults Study (CARDIA)." J Am Heart Assoc 5(10).

Organ, C. L., H. Otsuka, S. Bhushan, Z. Wang, J. Bradley, R. Trivedi, D. J. Polhemus, W. H. Tang, Y. Wu, S. L. Hazen and D. J. Lefer (2016). "Choline Diet and Its Gut Microbe-Derived Metabolite, Trimethylamine N-Oxide, Exacerbate Pressure Overload-Induced Heart Failure." Circ Heart Fail 9(1): e002314.

Reed, D. R. and A. Knaapila (2010). Genetics of taste and smell: poisons and pleasures. Progress in molecular biology and translational science, Elsevier. 94: 213-240.

Schmedes, M., C. Balderas, E. K. Aadland, H. Jacques, C. Lavigne, I. E. Graff, O. Eng, A. Holthe, G. Mellgren, J. F. Young, U. K. Sundekilde, B. Liaset and H. C. Bertram (2018). "The Effect of Lean-Seafood and Non-Seafood Diets on Fasting and Postprandial Serum Metabolites and Lipid Species: Results from a Randomized Crossover Intervention Study in Healthy Adults." Nutrients 10(5): 598.

Senthong, V., Z. Wang, Y. Fan, Y. Wu, S. L. Hazen and W. H. Tang (2016). "Trimethylamine N-Oxide and Mortality Risk in Patients With Peripheral Artery Disease." J Am Heart Assoc 5(10).

Shafi, T., N. R. Powe, T. W. Meyer, S. Hwang, X. Hai, M. L. Melamed, T. Banerjee, J. Coresh and T. H. Hostetter (2017). "Trimethylamine N-Oxide and Cardiovascular Events in Hemodialysis Patients." J Am Soc Nephrol 28(1): 321-331.

Tang, W. H., Z. Wang, B. S. Levison, R. A. Koeth, E. B. Britt, X. Fu, Y. Wu and S. L. Hazen (2013). "Intestinal microbial metabolism of phosphatidylcholine and cardiovascular risk." N Engl J Med 368(17): 1575-1584.

Troseid, M., T. Ueland, J. R. Hov, A. Svardal, I. Gregersen, C. P. Dahl, S. Aakhus, E. Gude, B. Bjorndal, B. Halvorsen, T. H. Karlsen, P. Aukrust, L. Gullestad, R. K. Berge and A. Yndestad (2015). "Microbiota-dependent metabolite trimethylamine-N-oxide is associated with disease severity and survival of patients with chronic heart failure." J Intern Med 277(6): 717-726.

Vigorita, M., S. Cozzolino, R. Oliva, G. Graziano and P. Del Vecchio (2018). "Counteraction ability of TMAO toward different denaturing agents." Biopolymers 109(10): e23104.

WHO. Global Health Observatory data – life tables: Japan. http://apps.who.int/gho/data/?theme=main&vid=60820. 2018, A. J., 2018.

WHO. Global Health Observatory data – life tables: USA. http://apps.who.int/gho/data/?theme=main&vid=61780. 2018, A. J., 2018.

Widmer, R. J., A. J. Flammer, L. O. Lerman and A. Lerman (2015). "The Mediterranean diet, its components, and cardiovascular disease." The American journal of medicine 128(3): 229-238.

Yancey, P. H., B. Speers-Roesch, S. Atchinson, J. D. Reist, A. R. Majewski and J. R. Treberg (2018). "Osmolyte Adjustments as a Pressure Adaptation in Deep-Sea Chondrichthyan Fishes: An Intraspecific Test in Arctic Skates (Amblyraja hyperborea) along a Depth Gradient." Physiol Biochem Zool 91(2): 788-796.

Yin, J., S. X. Liao, Y. He, S. Wang, G. H. Xia, F. T. Liu, J. J. Zhu, C. You, Q. Chen, L. Zhou, S. Y. Pan and H. W. Zhou (2015). "Dysbiosis of Gut Microbiota With Reduced Trimethylamine-N-Oxide Level in Patients With Large-Artery Atherosclerotic Stroke or Transient Ischemic Attack." J Am Heart Assoc 4(11).

Zou, Q., B. J. Bennion, V. Daggett and K. P. J. J. o. t. A. C. S. Murphy (2002). "The molecular mechanism of stabilization of proteins by TMAO and its ability to counteract the effects of urea."  124(7): 1192-1202.

Round 2

Reviewer 1 Report

no further comments

Reviewer 2 Report

Thank you for reading our peer review and giving an appropriate answer.

I think that there are no major problems to revise.